# An Epidemiological Assessment of *Cryptosporidium* and *Giardia* spp. Infection in Pet Animals from Taiwan

**DOI:** 10.3390/ani13213373

**Published:** 2023-10-31

**Authors:** Chia-Hui Hsu, Chi Liang, Shi-Chien Chi, Kuan-Ju Lee, Chung-Hsi Chou, Chen-Si Lin, Wen-Yuan Yang

**Affiliations:** 1Center for Animal Health and Food Safety, College of Veterinary Medicine, University of Minnesota, Saint Paul, MN 55108, USA; hsu00124@umn.edu; 2Department of Veterinary Medicine, National Taiwan University, Taipei 10617, Taiwan; r10629031@ntu.edu.tw (C.L.); b07606003@ntu.edu.tw (S.-C.C.); b06609062@ntu.edu.tw (K.-J.L.); cchou@ntu.edu.tw (C.-H.C.)

**Keywords:** *Cryptosporidium*, SSU-rRNA, *β*-giardin, glutamate dehydrogenase, *Giardia duodenalis*, nested polymerase chain reaction, zoonoses, molecular epidemiology, assemblages

## Abstract

**Simple Summary:**

*Cryptosporidium* spp. and *Giardia duodenalis*, enteric protozoan pathogens affecting humans and animals, elicit substantial global public concern. This study conducted in Taiwan sought to determine the prevalence and co-infection rates of Cryptosporidium spp. and *G. duodenalis* in dogs and cats. The investigation encompassed an analysis of infection rates and associated risk factors within the surveyed population. Predominantly identified species were *C. canis* and *C. felis*, aligning with canine-specific assemblages C and D. In contrast, the infrequent presence of human-specific assemblage A was noted in Giardia-positive samples. Phylogenetic analysis alluded to the potential for zoonotic transmission originating from domesticated animals. This underscores the role of pets as possible reservoirs for the transmission of cryptosporidiosis and giardiasis to humans in Taiwan.

**Abstract:**

*Cryptosporidium* spp. and *Giardia duodenalis* are enteric protozoan pathogens in humans. and animals. Companion animals infected with zoonotic species/assemblages are a matter of major public concern around the world. The objectives of the present study were to determine the prevalences of *Cryptosporidium* spp. and *G. duodenalis* infections and their co-infection statuses in dogs and cats living in Taiwan and to identify the species and assemblages. Fecal samples were collected from local animal shelters (*n* = 285) and a veterinary hospital (*n* = 108). Nested polymerase chain reaction (PCR) was performed using the SSU-rRNA, *β*-giardin, and glutamate dehydrogenase genes for *Cryptosporidium* spp. and *G. duodenalis*, respectively. Results showed that the overall prevalences of *Cryptosporidium* and *G. duodenalis* were 7.38% (29/393) and 10.69% (42/393). In addition, co-infection was detected in 1.02% (4/393) of all samples. Sample source, clinical sign, and breed may be risk factors that influence the infection rate. In *Cryptosporidium*-positive samples, *C. canis* and *C. felis* were detected most frequently. Although the canine-specific assemblages C and D (37/42) were dominant, the zoonotic human-specific assemblage A (1/42) was also found in *Giardia*-positive samples. Phylogenetic analysis revealed that most positive samples belonged to host-specific subtypes/assemblages, while some *Cryptosporidium* or *Giardia*-positive samples could be zoonotic. The findings suggested that pet animals could be a cause of zoonotic transmission, causing human cryptosporidiosis and giardiasis in Taiwan.

## 1. Introduction

Cryptosporidiosis and giardiasis are emerging infectious diseases caused by *Cryptosporidium* spp. and *Giardia intestinalis* (syn. *G. duodenalis* or *G. lamblia*), protozoan intestinal parasites which are of significant public health concern worldwide because infections are transmitted primarily through zoonotic, water-borne, and foodborne routes [1,2]. Human beings, companion animals, birds, domestic livestock, and a wide range of vertebrates all potentially contribute *Cryptosporidium* or *Giardia* spp. to the environment [3]. The main clinical presentation of cryptosporidiosis and giardiasis in humans and animals manifests as abdominal pain, watery diarrhea, dehydration, malabsorption, and wasting [4]. Generally, *Cryptosporidium* infections in immunocompetent people are mild and self-limiting. However, the lives of young children, the elderly and the immunocompromised can be threatened by infections [1,5]. In human beings, although *Cryptosporidium* and Giardia infections are only occasionally found, such as in immunocompromised patients, they still represent a significant health concern.

Among the approximately 40 recognized *Cryptosporidium* species, *C. hominis*, *C. parvum*, *C. meleagridis*, *C. canis*, and *C. felis* are the most prevalent in humans [6]. Similarly, within the eight common genotypes (A to H) of *G. duodenalis*, only assemblages A and B pose significant human health risks [7]. *Cryptosporidium* spp. and *G. duodenalis* are frequently found in dogs and cats worldwide [8,9]. *Cryptosporidium canis* and *C. felis* are the primary *Cryptosporidium* species in dogs and cats, respectively. However, occasional detections of *C. hominis*, *C. parvum*, *C. muris*, and *C. ubiquitum* have been reported in these animals [10,11,12,13,14]. Similarly, dog-adapted assemblages C and D and the cat-adapted assemblage F are the dominant *G. duodenalis* genotypes in these animals. However, zoonotic assemblages A and B have been identified in some studies [7,15].

Companion animals may play an essential role in the transmission routes of *Cryptosporidiosis* and *Giardiasis* [16]. Animal shelters are environments that facilitate the spread of gastrointestinal protozoan parasitic disease to incoming animals and shelter staff due to overcrowding and the multiple stressors of the animals. In addition to the animals, shelter staff and visitors are also considered *vulnerable to* exposure to zoonotic diseases [17]. Since both *Cryptosporidiosis and Giardiasis* are transmitted to humans through the fecal–oral route or contaminated food and water, infections can be more prevalent in crowded conditions if the shelter settings are unsanitary. In this research, we focused on sources in local animal shelters and household pets and analyzed the associations with risk factors. Hence, in this study, we collected stool specimens from one veterinary teaching hospital and three local animal shelters in Taiwan.

Traditionally, the diagnosis of *Cryptosporidium* and *Giardia* relies on detecting specific oocyst/cyst characteristics in stool samples, typically using acid-fast staining [18] or immunofluorescence assays with antibodies. However, these traditional methods cannot distinguish between species based on morphology or host occurrence. To address this issue and enable quick and accurate identification of these protozoan parasites, there is a need for novel, rapid, and discriminatory analytical methods applicable in both developed and developing regions, suitable for clinical and environmental samples.

Diagnosing *Cryptosporidium* and *Giardia* with the traditional visual microscopic method is difficult due to the inability to differentiate between different species using morphology and/or host occurrence. Instead, a molecular technique such as polymerase chain reaction (PCR) should be used to characterize *Cryptosporidium* and *Giardia* in feces or environmental samples [19]. Compared to the traditional microscopic method, nested PCR is more sensitive and reliable, so it is recommended for such detection and genotyping [20].

Although some surveys of *Cryptosporidium* oocysts and *Giardia* cysts have been conducted in livestock drinking water in Taiwan over the past decade, references or reports pertaining to *Cryptosporidium* and *Giardia* infection in small animal shelters are relatively limited. Thus, in this study, we performed the first cross-sectional investigation from different sources to offer a molecular assessment for future shelter animal medicine research. The primary aim of this research was to determine and elucidate the prevalences of *Cryptosporidium* and *Giardia* infection and their co-infection status in companion animals in Taiwan. The resulting information could serve as baseline data here. 

## 2. Materials and Methods

### 2.1. Study Design

In 2020, from February 1 to December 31, fresh canine and feline stool specimens were randomly collected from three local animal shelters (TW-TPE, TW-TYN, and TW-TTT; abbreviations of sampling sites were based on assigned country code) and one veterinary teaching hospital in Taiwan. The majority of the stool specimens were specifically recruited for the study, while the rest were obtained from routine diagnostics at the Section of Clinical Pathology, National Taiwan University Veterinary Hospital (NTUVH), Taipei. Among the 393 canine and feline specimens, 285 were from 3 local animal shelters, and 108 were from NTUVH. All the specimens were collected from live animals, and the collection processes were approved and followed local government regulations on animal protection. The study was approved by the Institutional Animal Care and Use Committee of National Taiwan University, Taipei, Taiwan (protocol code: IACUC No. NTU107-EL-00200).

### 2.2. Sample Collection

Each specimen (0.5–1 g) was collected from the ground immediately after animal defecation, placed into a 2 mL microcentrifuge tube, and then mixed with an appropriate amount of distilled deionized water (DDW). The properties of each specimen and the breed, age, gender, and past history of the animal were recorded in detail if possible. After several rounds of mechanical agitation, aliquots of the stool specimen were exposed to 5 freeze–thaw cycles, as described previously [21]. The processed specimens were stored at −20 °C, and DNA was extracted within 1 week after collection.

### 2.3. DNA Extraction

After the freeze–thaw cycles, total genomic DNA (gDNA) was isolated from the feces sample using the NautiaZ Stool DNA Extraction Mini Kit (Nautia Gene, Taipei, Taiwan) and eluted according to the manufacturer’s instructions. DNA concentration and purity were measured with a spectrophotometer (Eppendorf AG, Hamburg, Germany). The isolated gDNA samples were used immediately or stored at −20 °C for up to 1 month prior to screening.

### 2.4. Nested Polymerase Chain Reaction (PCR) Amplification

*Cryptosporidium* spp. was detected via nested-PCR amplification of the small subunit (SSU) rRNA gene. The DNA of *G. intestinalis* was amplified, targeting *β*-giardin (BG) and glutamate dehydrogenase (GDH) genes. As seen in Table 1, nested-PCR primer pairs were adopted in PCR amplification reactions in the study. A positive control (*Cryptosporidium* or *Giardia* DNA) and a negative control (distilled water) were run with every PCR batch. After nested-PCR amplification, 10 µL of each PCR product was electrophoresed by 2% agarose gel at 100 V for 30 min to identify the size of the products. PCR-positive products were purified using the PCR Clean-Up and Gel Extraction Kit (GeneDirex, Las Vegas, NV, USA), following the manufacturer’s instructions.

To calculate the estimated prevalence rate, we count the number of positive samples and divide it by the total number of samples tested using nested PCR. The overall prevalence rate is determined by dividing the number of positive samples by the total number of companion animal fecal samples that are being tested, irrespective of whether they are dogs or cats. We also calculated stratum-specific prevalence rates for dogs and cats separately.

### 2.5. Phylogenetic Analysis

To examine the genetic relationships among *Cryptosporidium* spp. and *Giardia duodenalis* assemblages in canines and felines, all positive secondary PCR amplicons (∼840 bp for *Cryptosporidium*; ~510 bp for BG in *Giardia*; ~530 bp for GDH in *Giardia*) were sent for nucleotide sequencing at Mission Biotech (Taipei, Taiwan). The results were compared with corresponding sequences on the National Center for Biotechnology Information (NCBI) website using the BLAST^®^ program (http://blast.ncbi.nlm.nih.gov/; accessed during 1 March~30 September 2020). A phylogenetic tree was constructed using the MEGA-X program with neighbor-joining (NJ) analysis of the SSU rRNA and BG gene sequences.

### 2.6. Software and Statistical Analysis

Geological maps were illustrated in the software QGIS 3.10.14. Statistical analyses were performed in Python 3.7 scipy.stats modules. Pearson’s chi-squared (χ^2^) test and Fisher’s exact tests were performed to test for any significant differences in location, gender, age, breed, and clinical signs between the two groups of canine and feline populations. Results were considered statistically significant at *p* < 0.05.

## 3. Results

### 3.1. Prevalence of Cryptosporidium and Giardia Infection and Co-Infection Pattern in Companion Animals

A total number of 393 individual stool samples were obtained for this study. Of them, 27.4% (108/393) of the specimens were collected from the veterinary hospital (TW-NTU) and 72.5% (285/393) from local shelters (TW-TPE, TW-TYN, and TW-TTT) (Figure 1 and Figure 2A). Of the canine fecal samples, 34.8% were from the veterinary hospital, and 65.2% were from local shelters. Of the feline samples, the percentages collected from the hospital and shelters were 14.7% and 85.3%, respectively. The overall prevalence rates of *Cryptosporidium* spp. and *G. duodenalis* by nested PCR were 7.38% (29/393) and 10.69% (42/393) (Figure 2B). In this study, the estimated prevalence rates of *Cryptosporidium* infection in dogs and cats were 8.4% (21/250) and 5.6% (8/143), while those of *G. duodenalis* infection in dogs and cats were 12.4% (31/250) and 7.69% (11/143), respectively (Table 2). Moreover, co-infection with both pathogens was detected in 1.02% (4/393) of all companion animal samples (Figure 2C,D). For identifying the specific assemblages of *G. duodenalis*, for example, 16 canine samples tested positive in Source-1 for BG, but only 8 samples for C and 6 for D were identified by genotyping. The remaining samples were not identified with specific assemblages. (Table 3).

### 3.2. Risk Factor Analysis

The univariable analyses of risk factors associated with positive detection results are shown in Table 4 and Table 5 for the two parasitic pathogens in canines and felines. Variables with *p* values < 0.05 were regarded as associated with infection. Animal gender, fecal source, breed, and clinical signs were evaluated as possible variables. Only gender was found to be not significantly related to both *Cryptosporidium* and *G. duodenalis* infections. The rates of *G. duodenalis* infection in canines and felines were higher in those from the local animal shelters than in those from the veterinary hospital. The probability of being infected by *G. duodenalis* was more than five times higher (OR = 5.8, 95% CI = 1.7–19.7, *p* = 0.0048) for the animal shelter source than for the veterinary hospital source. Dogs presenting diarrhea had a 2.9 times higher (OR = 2.9, 95% CI = 1.0–8.0, *p* = 0.0439) probability of *Cryptosporidium* infection. In contrast, cats, whether displaying noticeable clinical signs or not, were not linked to the prevalence of either disease. Interestingly, compared to mixed-breed felines, purebred cats were 9.5 times (OR = 9.5 95% CI = 1.9–47.1, *p* = 0.0058) more likely to be *Cryptosporidium*-positive. However, mixed-breed dogs were more likely to be *Giardia*-positive than purebred dogs (*p* = 0.0406).

### 3.3. Phylogenetic Analysis

Nested PCR and DNA sequencing results revealed that *Cryptosporidium* infection was detected in 7.38% of all fecal samples, whereas *Giardia* infection was detected in 10.69%. After DNA sequencing, the results showed that the species of *Cryptosporidium* were mainly *C. canis*, *C. felis*, *and C. meleagridis* (Figure 3A). Sequencing analysis of the SSU rRNA gene demonstrated that 72.4% (21/29) of the *Cryptosporidium*-positive samples shared 89–100% homology with the sequence of *C. canis (accession numbers:* AB210854.1, AF112576.1, KF516543.1, and MT329018.1) retrieved from the GenBank database. Within the nucleotide sequences of *Giardia*-positive samples isolated from companion animals, 61.9% (26/42) were successfully characterized as some specific assemblages, with most being assemblage C (17/26), followed by assemblage D (8/26) and assemblage A (1/26). Other parts of isolated *Giardia*-positive samples still presented as *β*-giardin-positive but matched as partial codons (Figure 3B). The most dominant species/assemblages of *Cryptosporidium*-/*Giardia*-positive samples were *C. canis* (*SSU-rRNA*; GenBank accession number AB210864.1, sequence homology 99–100%, *n* = 17) and *Giardia* canine-specific assemblage C (*β*-giardin; GenBank accession number LC437428.1, sequence homology 99–100%, *n* = 13).

## 4. Discussion

Our data showed that the overall *Cryptosporidium* infection rate was 7.38%. The prevalence of *Giardia* infection in companion animals identified at different genetic loci (*bg* and *gdh*) were 9.16% (36/393) and 3.56% (14/393), respectively. After the integration of positive samples of *Giardia* at the two loci, an overall 10.69% (42/393) prevalence was detected. A systemic review reported prevalence rates of *Giardia* of around 15.2% in dogs and 12% in cats worldwide [8]. In comparison, recent regional surveys to determine the prevalence of *Giardia* infection by ELISA antigen test or PCR have reported rates of 18.7% in Japan [25], 11.2% in South Korea [26], and 11% in China [27]. Our findings generally follow a pattern similar to those in surveys previously carried out in East Asia and worldwide.

The most noteworthy symptom in companion animals infected with protozoan parasites is diarrhea. According to our risk factor analysis, the risk of being positive for *Cryptosporidium* spp. was 2.9-fold higher in the diarrhea canine group than in the no-obvious-clinical-sign canine group. Meanwhile, our study also indicated that *Cryptosporidium* in felines and *Giardia* in both companion animals were not associated with diarrhea clinical signs. That is, although *Cryptosporidium* was confirmed in diarrhea cases, it was also commonly found in normal feces from patients without clinical signs. Both pathogens were detected regardless of companion animal diarrhea status. The detection of both endoparasites in normal feces suggests that the possibility of contamination with (oo)cysts in an entire population cannot be ruled out. This finding is important and has public health significance because pet owners and clinical veterinarians may neglect the possibility of infection of *Cryptosporidium* or *Giardia* due to their clinical presentation. These results should be considered by small-animal veterinarians in differential diagnosis.

Among the known *Cryptosporidium* species, human cryptosporidiosis is mostly caused by five species: *C. parvum*, *C. hominis*, *C. meleagridis*, *C. felis*, and *C. canis* [6,28,29]. In some countries, *C. meleagridis* may be as common as *C. parvum* [28,30,31]. In our findings, we detected the species *C. meleagridis* (*n* = 1) and *C. canis* (*n* = 18) in dogs, while *C. felis* (*n* = 5) and *C. canis* (*n* = 3) were identified in cats. For *Giardia,* 88.1% were classified as assemblage C or D. Moreover, one feline sample was successfully classified as *Giardia* assemblage A (*β*-giardin; GenBank accession number LC437420.1, sequence homology 100%). The three *Cryptosporidium* species and *Giardia* spp. genotype A recognized by DNA sequencing is known as zoonotic endoparasites.

Although human cryptosporidiosis infection through anthroponotic transmission is mainly caused by *C. parvum* and *C. hominis, which are* waterborne, *C. canis* and *C. felis* may infect humans through intrafamilial transmission by companion animals [32]. Thus, the horizontal infection of *Cryptosporidiosis* and *Giardiasis* in local animal shelters in Taiwan may present a public health threat to humans. As noted in our previous discussion, there is no significant difference between clinical signs in *Giardia*-positive pets. Therefore, we may neglect the potential risk when contacting clinically healthy companion animals.

The hosts of *Cryptosporidium* are virtually all vertebrate animal species. In Taiwan, the shelter facilities vary greatly by size, capacity, budget, and the number of governmental staff. These shelters offer temporary housing for not only stray dogs and feral cats but also a wide range of animals brought in by animal rescue services in local regions. Thus, wild birds and wildlife such as ferret badgers may be housed simultaneously in a high-density facility. The increased probability of exposure to contaminated water and food in animal shelters may explain the higher percentage of parasitic opportunistic infections. For example, birds are the natural host of *C. meleagridis* [33]. It is not surprising that the endoparasite may spread from birds to canines within the stressful environment of a local animal shelter.

*Giardia* cysts and *Cryptosporidium* oocysts can survive for months in environments with high humidity, low temperature, and low exposure to sunlight [34]. In Taiwan, the average monthly temperatures range from 17.9 °C in January to 30.9 °C in July, and the average annual relative humidity ranges from 73.0% to 80.6% [35]. Additionally, in the past two decades, *Cryptosporidium* oocysts and *Giardia* cysts have been documented in both drinking water for livestock and urban tap water in Taiwan [36,37]. One of the major features of giardiasis and cryptosporidiosis is their low infectious doses. Fewer than 10 cysts given orally could cause infection of *G. duodenalis,* while *Cryptosporidium* variably requires from 9 to 1024 oocysts [34,38]. The potential for contamination through drinking water should be investigated.

On the other hand, dogs and cats in animal shelters commonly come into contact with adopters and visitors. Although veterinarians regularly impose the disinfection process and exo-parasite chemical deworming program before the animals are admitted, the infection risk of endoparasites still exists. Moreover, once *Cryptosporidium* or *Giardia* (oo)cytes are transmitted in animal shelters, the horizontal transmission may spread dramatically because of oral–fecal infection within companion animals, owing to the narrow space and high-contact environment. Thus, to reduce the risk of zoonotic disease occurrence, fecal samples of pet animals should be routinely submitted for parasitic diagnostic tests, and owners should be informed about the public health issues related to pet fecal pollution. In local animal shelters, expansion of the disinfection checkpoints to maintain the biosecurity of the animal shelters is warranted.

There are some limitations to this research. Firstly, given the intermittent shedding pattern of (oo)cytes of *Cryptosporidium* and *Giardia*, molecular examinations should not only be based on fecal samples so as to prevent underestimation of the prevalence of each parasite. Secondly, increases in the levels of *Cryptosporidium* spp. and *Giardia* infection in young companion animals have been documented previously [39,40]. The age of the companion animals is commonly included in risk factor analysis. Nevertheless, in our sampling reference, the medical records on age are relatively vague and incomplete, complicating the statistical analysis. Third, viral disease infections should be considered necessary in enteropathogenic co-infection in kennels and shelters. The co-infection pattern is not limited to dual infection. In cats in the United Kingdom, co-infection of feline panleukopenia virus, *Cryptosporidium* spp., and *Giardia* spp. has been reported with double or triple infections [41]. The concurrent presence of canine parvovirus type 2, *Cryptosporidium* spp., and *Giardia* spp. has also been noted in Brazil [42]. However, no routine examinations of gastrointestinal viral pathogens are required in Taiwan. An analysis of the necessity of such requirements may be the next step in our research. Given these limitations, the zoonotic infection findings should be interpreted cautiously when generalized to a larger population of companion animals.

## 5. Conclusions

The present study demonstrated that the most dominant *Cryptosporidium* and *Giardia* species/assemblage characteristics in companion animals were *C. canis*, *C. felis*, and *Giardia* assemblages C and D, respectively. This is the first cross-sectional molecular evaluation of animals derived from local animal shelters and household pets in Taiwan. Notably, *C. canis*, *C. felis*, *C. meleagridis*, and *Giardia* assemblage A have the potential for zoonotic transmission. These molecular results can serve as a baseline to assess the zoonotic potential of companion animals in Taiwan. Further epidemiological studies are necessary to understand the transmission better.

## Figures and Tables

**Figure 1 animals-13-03373-f001:**
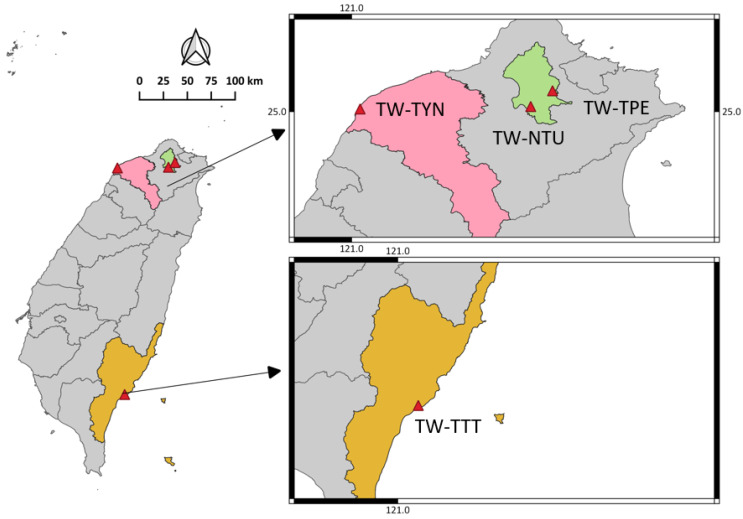
Map showing the sampling locations in Taiwan. Fresh canine and feline stool samples were collected in 2020 from three locations in Taiwan, including two urban areas (Taipei and Taoyuan) and a relatively rural area (Taitung). TW, Taiwan; TPE, Taipei; NTU, National Taiwan University; TYN, Taoyuan; TTT, Taitung.

**Figure 2 animals-13-03373-f002:**
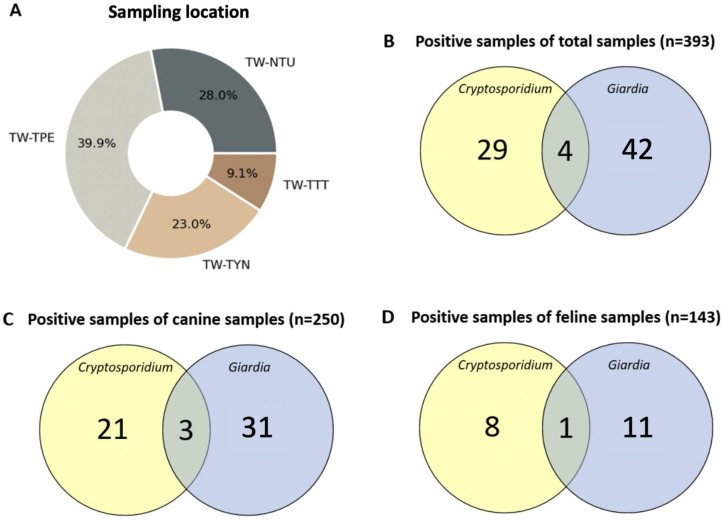
Percentages of the fecal samples from different locations, and the numbers of *Cryptosporidium* and *Giardia*-positive samples. (**A**) A pie chart of different sampling sources and sample sizes. (**B**–**D**) Venn diagrams showing the sample numbers of nested-PCR-positive results and co-infections of total, canine, and feline samples, respectively. TW, Taiwan; TPE, Taipei; NTU, National Taiwan University; TYN, Taoyuan; TTT, Taitung.

**Figure 3 animals-13-03373-f003:**
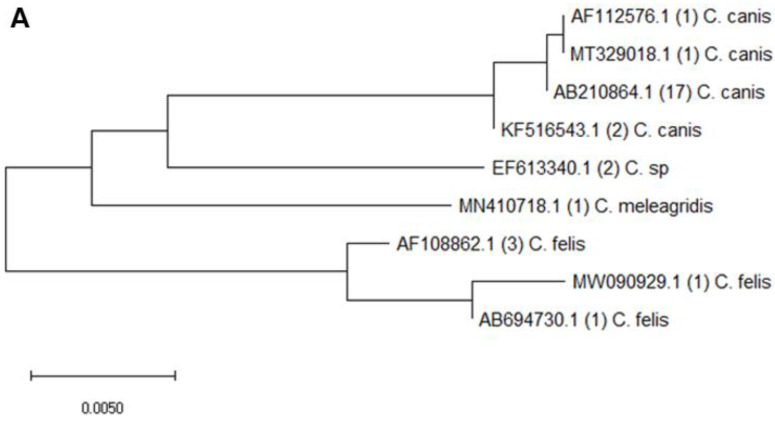
Phylogenetic analysis based on (**A**) SSU rRNA gene of *Cryptosporidium* and (**B**) BG gene of *Giardia*. The phylogenetic tree was constructed using MEGA software (1.0) by employing the neighbor-joining (NJ) method. The numbers in parentheses show the number of different species or assemblages. CDS, coding sequence.

**Table 1 animals-13-03373-t001:** Primers and annealing conditions of nested PCR for *Cryptosporidium* spp. SSU gene, *Giardia* BG gene, and *Giardia* GDH gene.

Primer Name	Primer Sequence (5’→3’)	Annealing (°C/s)	Product Size (bp)	Reference
Nested-PCR primers for *Cryptosporidium* SSU genes				
SSU-F1	TTCTAGAGCTAATACATGCG	55/45	1325	(Xiao et al., 1999) [22]
SSU-R1	CCCATTTCCTTCGAAACAGGA			
SSU-F2	GGAAGGGTTGTATTTATTAGATAAAG	55/45	840	
SSU-R2	CTCATAAGGTGCTGAAGGAGTA			
Nested-PCR primers for *Giardia* BG genes				
BG-F1	AAGCCCGACGACCTCACCCGCAGTGC	65/30	735	(MarcoLalle et al., 2005) [23]
BG-R1	GAGGCCGCCCTGGATCTTCGAGACGAC			
BG-F2	GAACGAACGAGATCGAGGTCCG	65/30	511	
BG-R2	CTCGACGAGCTTCGTGTT			
Nested-PCR primers for *Giardia* GDH genes				
GDH-F1	TTCCGTRTYCAGTACAACTC	57.8/30	754	(S.M.Cacciò et al., 2008) [24]
GDH-R1	ACCTCGTTCTGRGTGGCGCA			
GDH-F2	ATGACYGAGCTYCAGAGGCACGT	57.8/30	530	
GDH-R2	GTGGCGCARGGCATGATGCA			

**Table 2 animals-13-03373-t002:** Percentages and numbers of nested-PCR-positive canine and feline samples.

Category	No. Examined	Canine				No. Examined	Feline			
Pathogen		*Cryptosporidium* (%)		Giardia (%)			*Cryptosporidium* (%)		Giardia (%)	
Source-1 TW-TPE	82	9	11.0%	19	23.2%	76	4	5.3%	6	7.9%
Source-2 TW-TYN	55	2	3.6%	7	12.7%	36	1	2.8%	3	8.3%
Source-3 TW-TTT	26	3	11.5%	2	7.7%	10	0	0.0%	1	10.0%
Subtotal	163	14	8.6%	28	17.2%	122	5	4.1%	10	8.2%
Source-4 TW-NTU	87	7	8.0%	3	3.4%	21	3	14.3%	1	4.8%
Total	250	21	8.4%	31	12.4%	143	8	5.6%	11	7.7%

**Table 3 animals-13-03373-t003:** Percentages and numbers of *G. duodenalis* isolates of assemblage in canine and feline samples.

Category	No. Examined	Assemblages (n) of Canine					No. Examined	Assemblages (n) of Feline				
Pathogen		Overall	BG		GDH			Overall	BG		GDH	
Source-1 TW-TPE	82	19	16	C (8), D (6)	7	C (3), D (3)	76	6	6	A (1)	0	
Source-2 TW-TYN	55	7	6	C (5), D (1)	3	C (3)	36	3	3	C (2)	1	F (1)
Source-3 TW-TTT	26	2	1	D (1)	2	D (2)	10	1	0		1	D (1)
Source-4 TW-NTU	87	3	3	C (2)	0		21	1	1		0	
Total	250	31	26		12		143	11	10		2	

**Table 4 animals-13-03373-t004:** Risk factor analysis for canine cases.

Factor	*n*	*Cryptosporidium* spp.				*G. duodenalis*			
		No. Positive	OR	95%CI	*p*-Value	No. Positive	OR	95%CI	*p*-Value
Gender									
Female	113	11	1.4	(0.6–3.4)	0.49	12	0.7	(0.3–1.6)	0.44
Male	106	7	0.7	(0.3–1.7)	0.38	12	0.8	(0.4–1.8)	0.66
Unknown	31	3	1.2	(0.3–4.3)	0.78	7	2.4	(0.9–6.1)	0.07
Total	250	21				31			
Source									
Veterinary Hospital	87	7	1	(0.4–2.4)	0.88	3	0.2	(0.1–0.6)	0.0048 **
Animal Shelter	163	14	1.1	(0.4–2.8)		28	5.8	(1.7–19.7)	
Total	250	21				31			
Breed									
Purebred	64	4	0.7	(0.2–2.1)	0.47	3	0.3	(0.1–0.9)	0.0406 *
Mixed	186	17	1.5	(0.5–4.7)		28	3.6	(1.1–12.3)	
Total	250	21				31			
Clinical Sign									
Diarrhea	34	6	2.9	(1.0–8.0)	0.0439 *	7	2.1	(0.8–5.3)	0.13
No notable sign	216	15	0.3	(0.1–1.0)		24	0.5	(0.2–1.2)	
Total	250	21				31			

Note: * *p*-value < 0.05; ** *p*-value < 0.01.

**Table 5 animals-13-03373-t005:** Risk factor analysis for feline cases.

Factor	*n*	*Cryptosporidium* spp.				*G. duodenalis*			
		No. Positive	OR	95% CI	*p*-Value	No. Positive	OR	95% CI	*p*-Value
Gender									
Female	50	3	1.1	(0.3–4.9)	0.88	1	0.17	(0.0–1.4)	0.10
Male	46	4	2.2	(0.5–9.3)	0.28	4	1.22	(0.3–4.4)	0.76
Unknown	47	1	0.3	(0.0–2.3)	0.24	6	2.66	(0.8–9.2)	0.12
Total	143	8				11			
Source									
Veterinary Hospital	21	3	3.9	(0.9–17.7)	0.08	1	0.61	(0.1–5.0)	0.65
Animal Shelter	122	5	0.3	(0.1–1.2)		10	1.6393	(0.2–13.5)	
Total	143	8				11			
Breed									
Purebred	11	3	9.5	(1.9–47.1)	0.0058 **	0	0.4594	(0.0–8.3)	0.60
Mixed	132	5	0.1	(0.0–0.5)		11	2.177	(0.1–39.3)	
Total	143	8				11			
Clinical Sign									
Diarrhea	20	0	0.3	(0.0–6.0)	0.45	1	0.5947	(0.1–4.9)	0.63
No notable sign	123	8	3	(0.2–54.3)		10	1.6814	(0.2–13.9)	
Total	143	8				11			

Note: ** *p*-value < 0.01.

## Data Availability

The data in the present study are available from the corresponding author upon reasonable request.

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
