# Peer review of "An Epidemiological Assessment of Cryptosporidium and Giardia spp. Infection in Pet Animals from Taiwan"

_animals, 2023, doi:10.3390/ani13213373_

Round 1

Reviewer 1 Report

Comments and Suggestions for Authors

  The theme is interesting, but it has a lot of bugs. I listed some of them below. Due to these errors I suggest reject the manuscript.   

Pag. 1

Lines 14 y 16. The scientific names must be in italics.

Lines 18 y 19. The sentence “Predominantly identified species were C. canis and C. felis, aligning with canine-specific assemblages C and D, while the infrequent presence of human-specific assemblage A was noted in Giardia-positive samples”

It seems to me that the authors are not clear, is confusing. Please check the sentence. They only found a sample of Giardia from the assemblage A.

Line 24. Delete point

Line 37.  There was one sample with assemblage A, were not several assemblages

Line 42. The scientific names must be in italics.

Introduction

Page 2

Line 46. Giardiasis. It must be in lower case.

Lines 50 and 51. The sentence in not clear “Human beings, companion animals, birds, domestic livestock, and a wide range of vertebrates all potentially contribute Cryptosporidium or Giardia spp. to the environment”

Line 52.  Cryptosporidiosis and giardiasis with lower case

Line 65. Cryptosporidiosis and Giardiasis. It must be in lower case.  The scientific names must be in italics

Line 68.  Without italics

Material and methods

Page 3.

Table 1

Line 133 and 134. In Table 1 they put the reference of (Xiao et al.,1999); but in the reference section I did not find.

 If the reference is this: (Xiao lEscalante LYang CSulaiman IEscalante AAMontali RJFayer RLal AA. Phylogenetic analysis of Cryptosporidium parasites based on the small-subunit rRNA gene locus. Appl Environ Microbiol 1999 Apr;65(4):1578-83.doi: 10.1128/AEM.65.4.1578-1583.1999) then 1) there is an error in oligo R1 (CCCTAAT….) by reference  CCCATTT.  2) did you change it, if so, then you have to say that you made a modification and 3) is it another reference?

  In R2 they also have errors like in R1. The oligo does not match the reference

CTCATAAGGTGCTGAAGGAGTA by  reference ´5´-AAGGAGTAGGAACAACCTCCA-3

MarcoLalle et al.,  2005  and S.M. Caccio et al., 2008 articles are not in the references section.     

Results

Pag 5.

 Figure 2B. I found difference in results showed in Venn diagrams in relation with the page 4, lines 151-165.

Cryptosporidium 29 and Giardia 42. Please check the Venn diagram. Figure 2C parasites in canines 21+ 31= 52. Please check the Venn diagram.Table 2. Giardia with italics. Feline not fenine

Fig 3. Missing

Fig 4. Is Fig 3?

 Page 5

The results are unclear. The authors made the diagnosis using molecular biology techniques. In the case of Giardia, the data do not match because, description and table 2 are contradictory. 

1) In the manuscript (page 4, lines 162,163), they say that in dogs, they found 31 positive samples for Giardia, and in cats there were 11 positive samples, giving an average of 42 Giardia positive samples.

2) In Table 2, authors say that the dog samples using Betagiardin sequence found the assemblies: C= 15; D= 8 and with the GDH were C= 6 and D=5

In cats with Betagiardin were A= 1, C= 2, and with the GDH found D= 1 and F=1.

Question.

 Were both sequences used to genotype all Giardia samples? Were found assemblages confirmed with the two genes or did not?

Category 1 TW-TPE consisted of 82 samples, 19 genotyped. From 16 samples betagiardin genotyped, only gave results of 14 (Table 2). I wish to know what did happen to the first three samples (19-16= 3)? Also, if they found C=8 and D=6, what happened to the other two samples?

The most interesting question is how did they make the diagnosis of Giardia? Did the authors already know that the samples were Giardia positive?

In any case, Author wrote different results in lines 162 and 163. How many samples were Giardia positives, 39 or 42?

I found the same kind of errors in the Table 2.

Citation are not in concordance with the instruction for the authors.

Articles with the primers sequences were not in references section.  

Author Response

Response to the reviewer (R1)

Pag. 1

Lines 14 y 16. The scientific names must be in italics.

Response: Thank the reviewer for the comments. We have corrected all the scientific names in italics in this manuscript.

Lines 18 y 19. The sentence “Predominantly identified species were C. canis and C. felis, aligning with canine-specific assemblages C and D, while the infrequent presence of human-specific assemblage A was noted in Giardia-positive samples” It seems to me that the authors are not clear, is confusing. Please check the sentence. They only found a sample of Giardia from the assemblage A.

Response: Thank the reviewer’s question. Since only one canine case in this study was identified as human-specific assemblage A, this is considered an uncommon situation. The above explanation is the specific meaning that this sentence intends to convey.

Line 24. Delete point

Response: The point has been deleted.

Line 42. The scientific names must be in italics.

Response: Thank the reviewer for the comments. We have corrected all the scientific names in italics in this manuscript.

Introduction

Page 2

Line 46. Giardiasis. It must be in lower case.

Response: It is in lowercase in the revised manuscript.

Lines 50 and 51. The sentence in not clear “Human beings, companion animals, birds, domestic livestock, and a wide range of vertebrates all potentially contribute Cryptosporidium or Giardia spp. to the environment”

Response:

Line 52.  Cryptosporidiosis and giardiasis with lower case

Response: They are in lowercase in the revised manuscript.

Line 65. Cryptosporidiosis and Giardiasis. It must be in lower case.  The scientific names must be in italics

Line 68.  Without italics

Response: These typos are all corrected in the revised manuscript.

Material and methods 

Page 3.

Table 1

Line 133 and 134. In Table 1 they put the reference of (Xiao et al.,1999); but in the reference section I did not find.

 Response: The info. in Table 1 has been modified.

Results

Pag 5.

Figure 2B. I found difference in results showed in Venn diagrams in relation with the page 4, lines 151-165. Cryptosporidium 29 and Giardia 42. Please check the Venn diagram. Figure 2C parasites in canines 21+ 31= 52. Please check the Venn diagram. Table 2. Giardia with italics. Feline not fenine

Response: Thank you for your carefulness; the calculation error and typo have been corrected.

Fig 3. Missing

Fig 4. Is Fig 3?

Response: Thank you for the correction. There are only three figures in this manuscript.

Page 5

The results are unclear. The authors made the diagnosis using molecular biology techniques. In the case of Giardia, the data do not match because, description and table 2 are contradictory. 

  • In the manuscript (page 4, lines 162,163), they say that in dogs, they found 31 positive samples for Giardia, and in cats there were 11 positive samples, giving an average of 42 Giardia positive samples.

Response: The total detection of G. duodenalis in dogs is 31, while in cats, it is 11, resulting in a combined total of 42, which is correct.

2) In Table 2, authors say that the dog samples using Betagiardin sequence found the assemblies: C= 15; D= 8 and with the GDH were C= 6 and D=5

In cats with Betagiardin were A= 1, C= 2, and with the GDH found D= 1 and F=1.

Response: We apologize for any confusion regarding Table 2. To enhance the clarity of the results in the paper, we would like to provide additional information. In our study, we identified 31 samples in dogs that tested positive for G. duodenalis using nested PCR. To determine the specific G. duodenalis assemblages within these samples, we utilized two markers: BG and GDH. If a sample tested positive for BG or GDH, it was classified as positive for G. duodenalis.

Regarding the differentiation of specific BG and GDH assemblage groups, we conducted nucleotide sequencing on the positive PCR amplicons. However, it's noteworthy that not all 31 samples from dogs yielded identifiable assemblages for BG and GDH. Specifically, we were able to identify BG-specific assemblages in 23 out of the 31 positive samples (as indicated, C=15; D=8), while the overall BG genotype was observed in 26 out of 31 samples, including both identified and unidentified assemblages. Simultaneously, for GDH-specific assemblages (C or D), we identified a total of 11 out of 31 samples (C=6; D=5), while overall GDH genotype was observed in 12 out of 31 samples. Similar interpretations were applied in feline samples as well.

We hope this clarification aids in a better understanding of our results."

Question. 

Were both sequences used to genotype all Giardia samples? Were found assemblages confirmed with the two genes or did not?

Category 1 TW-TPE consisted of 82 samples, 19 genotyped. From 16 samples betagiardin genotyped, only gave results of 14 (Table 2). I wish to know what did happen to the first three samples (19-16= 3)? Also, if they found C=8 and D=6, what happened to the other two samples?

Response: The correct interpretation is as follows: We identified 19 samples that tested positive for G. duodenalis (either BG or GDH positive) in canines. Among these 19 samples, 16 exhibited BG genotypes. Within this group of 16 BG Giardia samples, we could only identify the C-assemblage in 8 and the D-assemblage in 6 samples. For the remaining samples within the 16, we can only assert that they belong to the BG genotype. However, based on the sequencing results, it is not possible to determine their specific assemblages.

The most interesting question is how did they make the diagnosis of Giardia? Did the authors already know that the samples were Giardia positive?

In any case, Author wrote different results in lines 162 and 163. How many samples were Giardiapositives, 39 or 42?

Response: The Giardia-positive cases were confirmed by PCR described in this manuscript. The total number of Giardia-positive samples is 42, with 31 from canines and 11 from felines.

Reviewer 2 Report

Comments and Suggestions for Authors

This study contributes to widen knowledge on the distribution of two well known zoonotic pathogens in dogs and cats in Taiwan, where data on the potential role of companion animals as reservoir for zoonotic infetion was still not investigated.

The Authors have conducted a wide survey and the research effort made was noticeable, nevertheless there are some very major points to fix in the ms to make it really sound to scientific community and clearer to the readers.

In the attached file I summarize some major revisions which I consider essential before reconsideration of the ms.

Introduction

This section contains some very general and well-known assumption on Cryptosporidium sp. and Giardia. Sentences sound somewhere obvious or even incorrect (e.g. line 56-58)

l.59 – obvious and already stated; must go deeper in the specific topic and avoid repetitions of concepts throughout all the sections

l.60 horizontal – obvious and unnecessary

l.64 – “In addition..” just stated two lines above

l.68-72 the aim of the study goes at the end of the section, where it is actually already stated; synthesize and eliminate this part from here.

l.74-79: these assumptions are again very general; a wide number of tests have been described in literature for the detection of these two protozoans in fecal samples, each with pros and cons and field of application. If the Authors wish to compare sensitivity of the tests they chose for their study, this must be done adding more references which justify the choice of the nested pcr. What abobut real time assays? What about the different genetic markers?

l. 88-89: “we could then develop…” this is a future perspective maybe, and is not really developed in the article discussion, so eliminate this statement or, if this is one of the scope of the study, provide appropriate and more detailed comparison with the surveys made in other countries in the discussion (see also next lines).

No info are given in the introduction on species, assemblages and subassemblages of Crypto and Giardia whose transmission from animals (cats and dogs) to humans is really demonstrated. Stating that these two are “zoonotic” protozoan seems a very general statement. Moreover, which assemblages are mostly isolated from dogs and cats? Which of them have a greater host range and which are more species-specific?

M&M

Major points:

109 -there is some confusion about signalment details which have been collected and actually used in the statistical tests. Please compare with lines 147-148, 183-184

l.111 what is the difference between mechanical and physical? please explain better

124. rephrase; it sounds quite obvious that primers pairs were used in a nested pcr. Moreover, why the nested PCR was used for detecting Giardia duodenalis in the samples? If the Authors themselves state that some samples were positive only to one of the two genes targeted, shouldn’t there be a number of false negative samples?

Phylogenetic analysis: the constructed trees should be improved by including references from the public database, belonging to the different species/ assemblages. What about GDH – based tree and multilocus analysis?

Statistical analysis: Fisher test was used for only gender, location (I suppose provenance, hospital vs shelter?), and age, what about breed and clinical signs?? Please explain and choose which risk factors were considered. Age was finally considered or not? (see l.308-309)

Results:

l. 155-159 Concept repeated from M&M. Choose where to state the number of samples collected for each category, or, better, summarize in a table the results from shelters and hospital separately, and from dogs and cats. This would substitute fig. 2 and would be clearer, reporting not just numbers but also percentages.

How do you estimate prevalence rates? Include it in the M&M.

l.180: here and throughout the text: why canine and feline? are there any other species other then dogs and cats?

l-.183 the critical p-value has been already stated in M&M, no need to repeat it. The variables evaluated as risk factors d not overlap with those in M&M section. Define which risk factors were really evaluated and state it in the M&M section. Here, only results are to be listed.

l. 185 “companion gender” sounds a bit weird

l.193: “cats …were not associated to with a higher prevalence of both disease” rephrase

Is the difference in prevalence varying with breed somewhat related to the difference in distribution of purebred and non-purebreed animals in hospital versus sheters dogs and cats? Can this represent a bias? anyway I cannot find any discussion on this result which would be interesting to comment on.

Phylogenetic analysis

First lines are a repetition of results given at the beginning of the section (l.160). Here, only identification of the isolates should be given and correctly justified.

l.l.206: “mainly”: what about the others? Are there any other species of Cryptosporidium among your isolates? Moreover: on which basis you define them as belonging to these species? Only for C. canis the BLAST analysis showed 100% identity with other isolates in GenBank, but what about the other species? How were they identified at species level? the paragraph should be rewritten including results of the blast analysis (also in the title) for all isolates, stating which parasite species from which host species and number for each (or at least include the details in a table, not in the tree). The meaning of the last four lines of the paragraph is confused to me…what do they add? What the linkage with the statements above??

Simialrly, the characterization of the assemblages of Giardia was conducted on which basis? blast analysis? Phylogenetic analysis? On which gene? Figure 3b is not existent. What about GDH gene? Didi you consider to make a multilocus analysis?

Human-specific assemblage is incorrect. The assemblage has wide host range across different animal families. Similarly, assemblage C is not “specific” to dogs. Change this wording and explain which is the host range of the assemblages and which are shared between campanion anianls and humans in the introduction. In the discussion section you can furtherly comment on this.

Discussion

The paragraph contains many repetitions of results but a true interpretation of the results themselves is not given. First part regarding sensitivity of the tests is again the justification for the choice of the tests, which is already given in intro section. Moreover, referring to PCR is too generic for two pathogens for which a number of different molecular assays have been developed and employed, using different methods and gene markers; if you want to keep this concept on the high sensitivity of the tests chosen, this must be well justified and commented with appropriate references.

Second paragraph: repetition of results, with new details which should go in the results section (gdh positive samples only appear in here, if I’m not wrong). Prevalence values reported for comparison are not introduced in the composition of the sample: which animals did these surveys include? which was their origin (owned-stray)? “worldwide” is really too generic! If the Authors have conducted a comparison with surveys made all over the world in dogs and cats for the two pathogens, all references should be included. But comparison with regional reports is far more interesting e contestualized, nevertheless more details are to be included.

Third paragraph: in the first part the concept is repeated twice, and it is still a result: how do you explain this result? What do literature specifically say about the occurrence of diarrhea in affected dogs and cats and relatively to the two different pathogens? Stating “the most noteworthy symptom…is diarrhea” is again too generic.

Fourth paragraph: results list again and generic comments. See considerations above on the discussion of the different zoonotic and non zoonotic strains, trying to be more specific and reporting your results in comparison with appropriate references: just for example, which is the occurrence of Assemblage A, C and D in cats??? Are they common or not? Is C. meleagridis commonly reported in dogs and cats?

Here and throughout the text check the use of italics.

Remnant parts of the discussion do not add much to comment the results but add too generic concepts, so I suggest to remove them and comment more on the specific results of your survey, for example including a discussion on the risk factors: what about breed effects on prevalence rates in literature ? What about shelters and hospitals? what about gender? How do these factors change P% in other studies? I think the discussion and conclusion sections should be deeply reviewed before reconsideration.

Comments on the Quality of English Language

Language should be revised, not much from a formal point of view, but more on the correct use of scientific terms, which are too generic and sometimes unappropriate.

Author Response

Response to the reviewer

This study contributes to widen knowledge on the distribution of two well known zoonotic pathogens in dogs and cats in Taiwan, where data on the potential role of companion animals as reservoir for zoonotic infetion was still not investigated.

The Authors have conducted a wide survey and the research effort made was noticeable, nevertheless there are some very major points to fix in the ms to make it really sound to scientific community and clearer to the readers.

In the attached file I summarize some major revisions which I consider essential before reconsideration of the ms.

Introduction

This section contains some very general and well-known assumption on Cryptosporidium sp. and Giardia. Sentences sound somewhere obvious or even incorrect (e.g. line 56-58)

Response: Thank the significant comment from the reviewer. The sentence has been rephrased as follows:

“In human beings, although Cryptosporidium and Giardia infections are only occasionally found, such as in immunocompromised patients, they still represent a significant health concern.”. (L55-57)

l.59 – obvious and already stated; must go deeper in the specific topic and avoid repetitions of concepts throughout all the sections

Response: The similar sentence has been removed.

l.60 horizontal – obvious and unnecessary

Response: It has been deleted.

l.68-72 the aim of the study goes at the end of the section, where it is actually already stated; synthesize and eliminate this part from here.

Response: It has been deleted.

l.74-79: these assumptions are again very general; a wide number of tests have been described in literature for the detection of these two protozoans in fecal samples, each with pros and cons and field of application. If the Authors wish to compare sensitivity of the tests they chose for their study, this must be done adding more references which justify the choice of the nested pcr. What abobut real time assays? What about the different genetic markers?

Response: We have further compared the current methods for these protozoan detections. (L72-91)

  1. 88-89: “we could then develop…” this is a future perspective maybe, and is not really developed in the article discussion, so eliminate this statement or, if this is one of the scope of the study, provide appropriate and more detailed comparison with the surveys made in other countries in the discussion (see also next lines). 

Response: This sentence has been deleted.

No info are given in the introduction on species, assemblages and subassemblages of Crypto and Giardia whose transmission from animals (cats and dogs) to humans is really demonstrated. Stating that these two are “zoonotic” protozoan seems a very general statement. Moreover, which assemblages are mostly isolated from dogs and cats? Which of them have a greater host range and which are more species-specific? 

Response: We have written the species and assemblages of Crypto and Giardia in humans, dogs, and cats in the “introduction” of the revised manuscript. (P2)

Among the approximately 40 recognized Cryptosporidium species, C. hominis, C. parvum, C. meleagridis, C. canis, and C. felis are the most prevalent in humans [4]. Similarly, within the eight common genotypes (A to H) of G. duodenalis, only assemblages A and B pose significant human health risks [5]. Cryptosporidium spp. and G. duodenalis are frequently found in dogs and cats worldwide [6, 7]. Cryptosporidium canis and C. felis are the primary Cryptosporidium species in dogs and cats, respectively. However, occasional detections of C. hominis, C. parvum, C. muris, and C. ubiquitum have been reported in these animals [8-12]. Similarly, dog-adapted assemblages C and D and the cat-adapted assemblage F are the dominant G. duodenalis genotypes in these animals. However, zoonotic assemblages A and B have been identified in some studies [5, 13].”

M&M

Major points:

109 -there is some confusion about signalment details which have been collected and actually used in the statistical tests. Please compare with lines 147-148, 183-184

Response: The case profiling is carefully described in Table 2.

l.111 what is the difference between mechanical and physical? please explain better 

Response: This sentence has been modified as follows:

 “After several rounds of physical or mechanical agitation,…..”

  1. rephrase; it sounds quite obvious that primers pairs were used in a nested pcr. Moreover, why the nested PCR was used for detecting Giardia duodenalis in the samples? If the Authors themselves state that some samples were positive only to one of the two genes targeted, shouldn’t there be a number of false negative samples? 

Response: Addressing concerns about false-negatives and false-positives is crucial from a diagnostic perspective. To establish a more robust protocol, we should "ideally" use microscopy with direct fluorescent antibody testing (DFA) or Microscopy with trichrome staining as a standard for comparison with our nested-PCR results to determine sensitivity and specificity. However, in practice, the shortage of manpower and diagnostic equipment in animal shelters in Taiwan, along with a lack of sufficient trained veterinarians for diagnosis, makes implementing such procedures challenging. In this study, we were unable to accurately assess the percentage of samples that might have yielded false negatives due to these limitations. This is one of the reasons why we opted for nested PCR, instead of conventional PCR. Unlike traditional microscopic methods and conventional PCR, nested PCR offers better sensitivity (Ref 9), reducing the risk of false negatives.

Statistical analysis: Fisher test was used for only gender, location (I suppose provenance, hospital vs shelter?), and age, what about breed and clinical signs?? Please explain and choose which risk factors were considered. Age was finally considered or not? (see l.308-309)

Response: Thank you for the revision. We should make a revision on Line 176; “breed" and "clinical signs" are also examined using Fisher’s exact test because no large-sample approximations were applied in the 2x2 table.
    In Table 3, given the ORs values and 95% CIs, mixed breed and animal shelter are risk factors for canine to be tested G. duodenalis positive. 
    For the age question; unfortunately, we encountered limitations in our study related to the availability of comprehensive medical records, specifically regarding the age data obtained from the veterinary hospital. Some age data were missing or incomplete, preventing us from including precise age information in our analysis. Introducing incomplete or potentially biased age data was deemed unfeasible and, therefore, not incorporated into our study. In sum, age is not finally considered.

Results:

  1. 155-159 Concept repeated from M&M. Choose where to state the number of samples collected for each category, or, better, summarize in a table the results from shelters and hospital separately, and from dogs and cats. This would substitute fig. 2 and would be clearer, reporting not just numbers but also percentages. 

Response: The precise data were summarized in Table 2 and Table 3.

How do you estimate prevalence rates? Include it in the M&M.

Response: We included this information in the Method section in P4.

“To calculate the estimated prevalence rate, we count the number of positive samples and divide it by the total number of samples tested using nested-PCR. The overall prevalence rate is determined by dividing the number of positive samples by the total number of companion animal fecal samples that are being tested, irrespective of whether they are dogs or cats. We also calculated stratum-specific prevalence rates for dogs and cats separately.”

l.180: here and throughout the text: why canine and feline? are there any other species other then dogs and cats?

Response: We presented the data from canine and feline separately in this study.

  1. 185 “companion gender” sounds a bit weird

Response: “companion” has been deleted.

l.193: “cats …were not associated to with a higher prevalence of both disease” rephrase 

Response: This sentence has been modified as follows:

“In contrast, cats, whether displaying noticeable clinical signs or not, were not linked to the prevalence of either disease. (P7)”

Is the difference in prevalence varying with breed somewhat related to the difference in distribution of purebred and non-purebreed animals in hospital versus sheters dogs and cats? Can this represent a bias? anyway I cannot find any discussion on this result which would be interesting to comment on. 

Response: We have discussed the “location” factor in P11. If the pure- or nonpure breed animals could be the risk factor, our data in cats and dogs showed no apparent pattern. That is the reason why we did not put emphasis on this factor.

Simialrly, the characterization of the assemblages of Giardia was conducted on which basis? blast analysis? Phylogenetic analysis? On which gene? Figure 3b is not existent. What about GDH gene? Didi you consider to make a multilocus analysis?

Response: As described in M&M section 2.4 & 2.5 (P4), the assemblages of Giardia was identified by sequencing and blast analysis of β-giardin (BG) and glutamate dehydrogenase (GDH) genes.

Human-specific assemblage is incorrect. The assemblage has wide host range across different animal families. Similarly, assemblage C is not “specific” to dogs. Change this wording and explain which is the host range of the assemblages and which are shared between campanion anianls and humans in the introduction. In the discussion section you can furtherly comment on this.

Response: Thank you for the significant correction. We have revised these mistakes as follows: (P9)

“Within the nucleotide sequences of Giardia-positive isolated from companion animals, 72.2% (26/36) were successfully characterized as some specific assemblages, with most being assemblage C (17/26), followed by assemblage D (8/26) and assemblage A (1/26).”

    We also wrote the host range of the assemblages that are shared between companion animals and humans in the introduction. (P2)

Discussion

The paragraph contains many repetitions of results but a true interpretation of the results themselves is not given. First part regarding sensitivity of the tests is again the justification for the choice of the tests, which is already given in intro section. Moreover, referring to PCR is too generic for two pathogens for which a number of different molecular assays have been developed and employed, using different methods and gene markers; if you want to keep this concept on the high sensitivity of the tests chosen, this must be well justified and commented with appropriate references.

Response: This paragraph has been deleted.

Second paragraph: repetition of results, with new details which should go in the results section (gdh positive samples only appear in here, if I’m not wrong). Prevalence values reported for comparison are not introduced in the composition of the sample: which animals did these surveys include? which was their origin (owned-stray)? “worldwide” is really too generic! If the Authors have conducted a comparison with surveys made all over the world in dogs and cats for the two pathogens, all references should be included. But comparison with regional reports is far more interesting e contestualized, nevertheless more details are to be included.

Response: Thank the comment from the reviewer. However, we did compare our prevalence data with that from different countries and believe that this comparison is meaningful by providing the current data on companion animals in Taiwan.

Third paragraph: in the first part the concept is repeated twice, and it is still a result: how do you explain this result? What do literature specifically say about the occurrence of diarrhea in affected dogs and cats and relatively to the two different pathogens? Stating “the most noteworthy symptom…is diarrhea” is again too generic.

Response: Thank the comment from the reviewer. We intend to discuss the effects of cryptosporidium and Giardia infection on diarrhea symptoms in our sampling population.

Fourth paragraph: results list again and generic comments. See considerations above on the discussion of the different zoonotic and non zoonotic strains, trying to be more specific and reporting your results in comparison with appropriate references: just for example, which is the occurrence of Assemblage A, C and D in cats??? Are they common or not? Is C. meleagridis commonly reported in dogs and cats?

Response: We have compared our data with those from the previous findings.

Here and throughout the text check the use of italics. 

Response: Thank the reviewer for the comments. We have corrected all the scientific names in italics in this manuscript.

Comments on the Quality of English Language

Language should be revised, not much from a formal point of view, but more on the correct use of scientific terms, which are too generic and sometimes unappropriate.

Response: This article has been reviewed and edited by professional academic English editors.

Reviewer 3 Report

Comments and Suggestions for Authors

The manuscript reports the results of an epidemiological study on the occurrence of Giardia duodenal assemblages and Cryptosporidium species in dogs and cats from different environment in Taiwan. The paper is well written although does not add any new to the state of art

some minor issues are listed

Cryptosporidium in italics everywhere and Giardia, too, in key words

lines 56-57 and 64-65, 67-68, 208, 213, 259, 261, 264, 281..... - check italics character

line 75 - the size of Cryptosporidium oocysts and of Giardia cysts are not comparable. Please, reformulate the sentence

line 78 - change (oo)cysts in diagnostic stages

line 137 - G.duodenalis

line 306 - Cryptosporidium per extenso

Author Response

Response to the reviewer

The manuscript reports the results of an epidemiological study on the occurrence of Giardia duodenal assemblages and Cryptosporidium species in dogs and cats from different environments in Taiwan. The paper is well written although does not add any new to the state of art

some minor issues are listed

  1. Cryptosporidium in italics everywhere and Giardia, too, in key words

lines 56-57 and 64-65, 67-68, 208, 213, 259, 261, 264, 281..... - check italics character

Response: Thank the reviewer for the comments. We have corrected all the scientific names in italics in this manuscript.

  1. Line 75 - the size of Cryptosporidium oocysts and of Giardia cysts are not comparable. Please, reformulate the sentence

Response: The sentence has been rephrased as follows:

“Diagnosing Cryptosporidium and Giardia with the traditional visual microscopic method is difficult due to the inability to differentiate between different species using morphology and/or host occurrence.” (L74-76)

  1. line 78 - change (oo)cysts in diagnostic stages

Response: The sentence has been rephrased as follows:

“Instead, a molecular technique such as polymerase chain reaction (PCR) should be used to characterize Cryptosporidium and Giardia in feces or environmental samples”.

  1. line 137 - G.duodenalis

line 306 - Cryptosporidium per extenso

Response: These two names were corrected in the revised manuscript.

Reviewer 4 Report

Comments and Suggestions for Authors

The objective of this study is to provide information on animal fecal samples tested positive for Crypto and/or Giardia from various places in Taiwan.

The author here has shed light on the cross-sectional study on co-infection of Cryptosporidium and Giardia spp. in pet animals. The field data put together is impressive and adds on to the current knowledge in the field of parasitology emphasizing on the asymptomatic crypto positive pets. 

However, since the author has not provided any information on pets owner's history, symptoms or clinical evidence suggesting a case of vertical transmission, author is advised to restrict from mentioning it as one of the causes of human cryptosporidiosis and giardiasis in this epidemiological survey from animal samples in the summary and abstract sections.

Additionally, I would request the author to consider shortening the title of the manuscript to make it reader friendly and to broaden the scope of the paper from co-infection to Giardia and/or Crypto infection. This is because the of number of data points generated in the survey. A suggestive title that I would propose would be "An epidemiological assessment of Cryptosporidium and Giardia spp. infection in pet animals from Taiwan" Authors are requested to discuss on potential future plans of this data generated to broaden the scope of coinfection studies and potential zones of sample collection for the same in the future.

An interesting and informative take in this paper is seen in the risk factor analysis, where the infection is correlated with gender and breed of the animals. The author has ended the discussion on risk factor analysis reporting higher chances of mixed-breed dogs being Giardia-positive than purebred dogs. The author is requested to discuss on possible reasons for it being a biological phenomenon or due to a sampling bias from a limited pool. 

Comments on the Quality of English Language

Author is advised to revise minor grammatical and typographical errors that can be found in multiple lines of the manuscript. Additionally it is advised to either state reasons for italicising multiple lines in the discussion or review it if its due to typographical error. 

Author Response

Response to the reviewer

The objective of this study is to provide information on animal fecal samples tested positive for Crypto and/or Giardia from various places in Taiwan. 

The author here has shed light on the cross-sectional study on co-infection of Cryptosporidium and Giardia spp. in pet animals. The field data put together is impressive and adds on to the current knowledge in the field of parasitology emphasizing on the asymptomatic crypto positive pets. 

However, since the author has not provided any information on pets owner's history, symptoms or clinical evidence suggesting a case of vertical transmission, author is advised to restrict from mentioning it as one of the causes of human cryptosporidiosis and giardiasis in this epidemiological survey from animal samples in the summary and abstract sections.

Response: Thank the significant comment from the reviewer. In the revised manuscript, we have described the data we have obtained in a conservative manner in the summary and abstract.

Additionally, I would request the author to consider shortening the title of the manuscript to make it reader friendly and to broaden the scope of the paper from co-infection to Giardia and/or Crypto infection. This is because the of number of data points generated in the survey. A suggestive title that I would propose would be "An epidemiological assessment of Cryptosporidium and Giardia spp. infection in pet animals from Taiwan" Authors are requested to discuss on potential future plans of this data generated to broaden the scope of coinfection studies and potential zones of sample collection for the same in the future.

Response: The title has been changed as the reviewer suggested. The authors also plan the future works in the “discussion” section. (L342-346)

An interesting and informative take in this paper is seen in the risk factor analysis, where the infection is correlated with gender and breed of the animals. The author has ended the discussion on risk factor analysis reporting higher chances of mixed-breed dogs being Giardia-positive than purebred dogs. The author is requested to discuss on possible reasons for it being a biological phenomenon or due to a sampling bias from a limited pool. 

Response: We have discussed the “location” factor in P11. If the pure- or nonpure breed animals or gender could be the risk factors, our data in cats and dogs showed no apparent pattern. That is the reason why we did not put emphasis on these factors.

Comments on the Quality of English Language

Author is advised to revise minor grammatical and typographical errors that can be found in multiple lines of the manuscript. Additionally it is advised to either state reasons for italicising multiple lines in the discussion or review it if its due to typographical error. 

Response: This article has been reviewed and edited by professional academic English editors. All typos and italics have been corrected.

Round 2

Reviewer 1 Report

Comments and Suggestions for Authors

Comments are in the attached file

Author Response

Response to the Reviewer

Thank you for your valuable feedback, Reviewer 1.

Line 118-120. Due to what is described in these lines, the question arises again. Did you

already have the diagnosis of parasitosis?

In response to your query regarding lines 118-120, it is essential to clarify that the parasitosis diagnosis was not predetermined before our laboratory analysis. At NTUVH, there is no standardized protocol for routinely examining these two pathogens discussed in our study. We collected samples from both animal shelters and hospital sources, and to ensure consistency, we used a nested-PCR test for all the samples to determine their status as positive or negative in our research. To address your specific question, we did not possess any prior knowledge regarding whether the samples were positive or negative for parasitosis before conducting our laboratory analyses. The results were determined solely through our testing procedures.

Example of serious errors.

Line 188. There are errors in the number of samples (The authors say that there were 393 samples. From the hospital there were 108 samples and from the animal shelters there were 285 samples)

Hospital (108 samples): dog samples were 34.8% = 38 dogs. Cat samples were 14.7% = 16 cats. Total: 38 dogs + 16 cats = 54 samples. 54 samples missing Animal shelters (285 samples): Dog samples were 65.2% = 186 dogs. Cat samples were 85.3% = 243 cats. Total: 186 dogs + 243 cats = 429 samples. Why are there 144 samples left over?

The interpretation here is inaccurate because the "34.8%" means the ratio of 87 out of 250 (the number of dog samples from the hospital out of all the dog samples). We will discuss the question in detail in the next part.

On the other hand. Lines 193-196. The authors say that there were 250 samples from dogs and 143 samples from cats. There is contradiction in the data.

We appreciate your input and would like to address some potential areas of confusion to ensure the clarity of our paper. The misunderstanding appears to stem from the two levels of description we employ in our study, namely, location (either animal shelter or hospital) and species (cats or dogs). Here, we aim to provide a clear breakdown of the sample distribution to address your concerns.

In our study, we collected a total of 393 samples:

(1) 285 samples were obtained from animal shelters, and 108 samples were acquired from hospitals, totaling 393 samples.

(2) Within the entire dataset of 393 samples, 250 samples were of canine origin, and 143 were of feline origin, resulting in a sum of 393 samples.

(1)

Number

Detail

animal shelters

285

163 canine

122 feline

hospitals

108

87 canine

21 feline

Total

393

Let's examine this information in more detail:

For (1), focusing on animal shelters (a total of 285 cases), there were 163 canine samples and 122 feline samples, as indicated in Table 2. These numbers accurately sum up to the 285 samples retrieved from animal shelters.

For (1), looking at hospital samples (a total of 108 cases), there were 87 canine samples and 21 feline samples, again accurately totaling 108 samples.

(2)

Number

Detail

canine origin

250

163 from animal shelters (65.2%)

87 from hospitals (34.8%)

feline origin

143

122 from animal shelters (85.3%)

21 from hospitals (14.7%)

Total

393

For (2), within the 250 canine samples, 163 were from animal shelters (approximately 65.2%), and 87 were from hospitals (approximately 34.8%).

For (2), out of the 143 feline samples, 122 were from animal shelters (approximately 85.3%), and 21 were from hospitals (approximately 14.7%).

On the other hand. Lines 193-196. The authors say that there were 250 samples from dogs and 143 samples from cats. There is contradiction in the data.

Line 193-196, the number there is correct. 250 samples from dogs and 143 samples from cats.

The data is actually not contradictory.

The Venn diagram was wrong, thank you for the correction, it should be “42” in Giardia.

Correction should be like this.

The authors say there were 31 samples of dogs with Giardia. But in the Table, they give results from 23 genotyped samples. What happened to the other 8 samples? These results show that the authors already knew the diagnosis of giardiasis and do not say so in the manuscript. Regarding cats, 11 samples were positive for Giardia. I only see 5 genotyped samples. What happened to the other 6? These results show that the authors were already aware of the diagnosis of giardiasis.

We'd like to provide clarification on our procedure for identifying Giardia positivity in our samples. When a positive band was identified in the nested PCR, we categorized the sample as positive for Giardia. Subsequently, all positive secondary PCR amplifications of the sample were sent for nucleotide sequencing. These sequencing results were used to determine the specific genotype.

This process was successfully applied to 23 samples (in BG column) for which we determined the genotypes, and these genotypes are presented in Table 3. However, the remaining samples, while testing positive for Giardia, did not yield clear genotype results, preventing us from assigning specific genotypes to them in the table.

A similar situation was observed in the case of cat samples, where 11 samples were positive for Giardia. However, only 5 (1 A genotype, 2 C genotypes, 1 F genotype, and 1 D genotype) could be identified when compared to corresponding sequences in the NCBI Blast database.

These results show that the authors were already aware of the diagnosis of giardiasis.

Our study followed a transparent and systematic methodology, and the parasitosis diagnosis was not predetermined before our laboratory analysis. We conducted all procedures with complete transparency to report our results accurately. We appreciate your inquiry and remain committed to ensuring the clarity of our research process.

The prevalence results are not supported.

Examples of miscited references

Feng, Y. and L. Xiao, Zoonotic potential and molecular epidemiology of Giardia species and giardiasis. Clin Microbiol Rev, 2011. 405 24(1): p. 110-40.

Xiao, L., et al., Possible transmission of Cryptosporidium canis among children and a dog in a household. Journal of clinical microbiology, 2007. 45(6): p. 2014-2016

Li, J., et al., Genotype identification and prevalence of Giardia duodenalis in pet dogs of Guangzhou, Southern China. Veterinary parasitology, 2012. 188(3-4): p. 368-371

We have checked all references including the above 3 cases, and they are the correct citations supporting the prevalence results the authors intend to claim or discuss in this manuscript.   
